# Comparing the Clinical and Laboratory Features of Remitting Seronegative Symmetrical Synovitis with Pitting Edema and Seronegative Rheumatoid Arthritis: Stage 1

**DOI:** 10.3390/jcm10020340

**Published:** 2021-01-18

**Authors:** Misako Higashida-Konishi, Keisuke Izumi, Satoshi Hama, Hiroshi Takei, Hisaji Oshima, Yutaka Okano

**Affiliations:** 1Department of Connective Tissue Diseases, National Hospital Organization Tokyo Medical Center, Tokyo 1528902, Japan; izz@keio.jp (K.I.); shama@ntmc-hosp.jp (S.H.); htakei@ntmc-hosp.jp (H.T.); hoshimamac@mac.com (H.O.); yutakaokano@mac.com (Y.O.); 2Division of Rheumatology, Department of Internal Medicine, Keio University School of Medicine, Tokyo 1608582, Japan

**Keywords:** rheumatoid arthritis, synovitis, neoplasms, edema, inflammation

## Abstract

In seronegative arthritis with extremity edema, the differential diagnosis between remitting seronegative symmetrical synovitis with pitting edema syndrome (RS3PE) and seronegative rheumatoid arthritis (SNRA) is difficult. We compared the clinical characteristics of RS3PE and SNRA and those of such patients with and without malignancies. We retrospectively examined patients diagnosed with RS3PE (McCarty criteria) and SNRA at our hospital in 2007–2020. Malignancy was diagnosed within 2 years before or after RS3PE or SNRA diagnosis. Overall, 24 RS3PE and 124 SNRA patients were enrolled. The mean ages were 79.0 and 66.5 years, and men comprised 54.2% and 37.1% of RS3PE and SNRA patients, respectively. RS3PE patients had higher inflammation levels (*p* < 0.01) and more incidences of malignancy (*p* < 0.01). Matching for age and sex, RS3PE patients had higher inflammation levels (*p* < 0.01) and more incidences of malignancy (*p* = 0.02). Overall, odds ratios (ORs) for malignancy were higher for older age (OR 1.06, *p* = 0.04), male sex (OR 4.34, *p* = 0.02), RS3PE patients (OR 4.83, *p* = 0.01), and patients with extremity edema (OR 4.83, *p* = 0.01). RS3PE patients had higher inflammation levels and associated factors of malignancy than SNRA patients. Patients who are older, male, with extremity edema, or with RS3PE should be screened for malignancies.

## 1. Introduction

Remitting seronegative symmetrical synovitis with pitting edema (RS3PE) was first reported by McCarty et al. in 1985 [1]. It is characterized by pitting edema of the extremities, sudden onset of polyarthritis, seronegativity for rheumatoid factor (RF), excellent response to glucocorticoids, and the absence of radiologically evident erosions [1]. RS3PE mainly affects the joints of the extremities, especially the metacarpophalangeal (MCP) and proximal interphalangeal (PIP) phalanges, wrists, shoulders, elbows, knees, and ankles [2]. Although the pathophysiology of RS3PE remains unclear, vascular endothelial growth factor (VEGF) serum levels have been found to be elevated in patients with RS3PE [3]. The increase in vascular permeability by VEGF is thought to be responsible for the development of pitting edema of the dorsum of both hands and both feet in patients with RS3PE [3].

Initially, RS3PE was thought to be a type of older-onset rheumatoid arthritis (RA) [4] and was considered the same disease as seronegative RA and polymyalgia rheumatica (PMR) [5]. Subsequently, comparisons between PMR and RS3PE have been reported [6]. Kawashiri et al. reported the differences in musculoskeletal ultrasound findings of both hands between RS3PE and “seropositive” elderly onset RA; however, to our knowledge, no reports have compared the characteristics of RS3PE and “seronegative” RA [7].

RS3PE is often described as a paraneoplastic disease [8] and has been reported to have a high rate of malignancy development [9]. Paraneoplastic arthritis often presents as symmetrical polyarthritis, mainly affecting the wrist and fingers, and is often negative for RF and anti-cyclic citrullinated peptide antibody (ACPA) [10]. Early diagnosis of malignancy is clinically important because it improves survival. Therefore, examination for malignancy is necessary in such cases.

The primary aim of this study was to compare the clinical characteristics of RS3PE and seronegative RA and evaluate the frequency of concurrent malignancy. The secondary aim was to compare the clinical features with and without malignancies in patients with RS3PE and to compare the clinical features with and without malignancies in patients with seronegative RA.

## 2. Materials and Methods

### 2.1. Compliance with Ethical Standards

All procedures were performed in accordance with the ethical standards of the institutional and national research committees and the 1975/1983 Helsinki Declaration and its later amendments.

### 2.2. Study Design

This was a retrospective medical record study.

### 2.3. Patients

Medical records of consecutive patients diagnosed with RS3PE and seronegative RA at our hospital between 2007 and 2020 were retrospectively examined. Patients who were both ACPA- and RF-negative were included. Patients who met the criteria for both PMR and RS3PE were included in the RS3PE group and those who met the criteria for both PMR and seronegative RA were included in the seronegative RA group. PMR was diagnosed according to the 2012 European League Against Rheumatism/American College of Rheumatology (EULAR/ACR) Provisional Classification Criteria for PMR [11]. For patients diagnosed with PMR before 2012, we retrospectively reviewed whether they met the 2012 PMR classification criteria. Patients who met the criteria for both RA and RS3PE were diagnosed with RS3PE. However, those who had erosion were diagnosed with seronegative RA. We defined RS3PE and seronegative RA patients by excluding those who met the criteria for PMR as “pure RS3PE” and “pure seronegative RA.” Patients who met the criteria for both RA and PMR were diagnosed with seronegative RA. Patients with paraneoplastic polyarthritis were excluded from the group of patients with RS3PE or seronegative RA. Those with distal joint swelling that rapidly disappeared after tumor resection were diagnosed with paraneoplastic polyarthritis.

### 2.4. RS3PE Diagnosis

Patients were diagnosed with RS3PE when they met the McCarty et al. criteria [1]: (1) pitting edema of the dorsum of both hands and both feet, (2) sudden onset of polyarthritis, (3) seronegative for RF, and (4) no development of radiologically evident erosions.

### 2.5. Seronegative RA Diagnosis

Seronegative RA was diagnosed according to the 2010 EULAR/ACR criteria [12]. Patients who were first diagnosed with RS3PE or PMR and later diagnosed with seronegative RA were included in the seronegative RA group.

### 2.6. Clinical and Laboratory Features

We examined the affected joints and evaluated them for systemic signs and symptoms (temperature ≥ 38.0 °C, malaise or fatigue, weight loss, morning stiffness lasting at least 1 h, and edema). The affected joints were the shoulders, elbows, wrists, fingers (MCP and interphalangeal (IP)/PIP joints), hips, knees, ankles, and toes (MCP and IP/PIP joints). Edema was evaluated separately as edema of only hands, only feet, and of both limbs. We also measured the erythrocyte sedimentation rate (ESR) and the levels of C-reactive protein (CRP), hemoglobin (Hb), albumin (Alb), lactate dehydrogenase (LDH), and matrix metalloproteinase 3 (MMP-3). Smokers were defined as those who had a smoking history within 2 years before and after RS3PE or seronegative RA diagnosis. If there were evaluable examinations, ultrasound imaging, breast imaging, joint X-ray imaging, chest computed tomography (CT), abdominal CT, pelvic CT, positron emission tomography/CT, joint magnetic resonance imaging, upper and lower gastrointestinal endoscopy, gynecological examination, and pathological tests were performed.

### 2.7. Statistical Analysis

The first analysis was performed on clinical and laboratory features of patients with RS3PE and seronegative RA. The secondary analysis was performed on the above evaluations with matching for age and sex. All data were analyzed using JMP version 14.0 (SAS Institute, Cary, NC, USA). The third analysis was performed to compare the clinical features of patients with or without malignancy among patients with RS3PE or seronegative RA. Univariate analysis, Fisher’s exact test, and logistic regression analysis were applied to evaluate the associated factor of malignancy. A probability level less than 0.05 was used as the criterion of significance. Results that did not follow the Gaussian distribution were expressed as the median of the 25–75th percentile (interquartile range), and results that followed the Gaussian distribution were expressed as mean ± standard deviation.

## Data Availability

Not available.

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
