# Peer review of "Comparing the Clinical and Laboratory Features of Remitting Seronegative Symmetrical Synovitis with Pitting Edema and Seronegative Rheumatoid Arthritis: Stage 1"

_jcm, 2021, doi:10.3390/jcm10020340_

Round 1

Reviewer 1 Report

This is a cross-sectional study to evaluate differences in the clinical characteristics of RS3PE and SNRA, especially in the association of malignancies. It is a clinically interesting manuscript. The research question is appropriate, but the methodology used for the analyses needs to be modified.

Major comments

  1. The authors compared the clinical characteristics between the patients with RS3PE and SNRA. Propensity score matching was used to control for potential confounding. However, It is not clear what the main outcome or confounders. In addition, the sample size is too small to use PS matching. It’s enough to use age- and sex- matching, if it is needed.
  2. The third analysis is a kind of case-control analysis to identify if the RS3PE is a risk factor for the development of malignancies. In this analysis, malignancy is an outcome and it should have happened after the development of RS3PE or SNRA. However, the associating malignancy was defined as they were diagnosed within 2 years before or after RS3PE or SNRA diagnosis. The purpose of this study was to compare the clinical characteristics of RS3PE and SNRA, but their conclusion is strange. They concluded that patients who are older, male, with extremity edema, or with RS3PE should be screened for malignancies.

Author Response

Thank you for your comment.

Point 1:

The authors compared the clinical characteristics between the patients with RS3PE and SNRA. Propensity score matching was used to control for potential confounding. However, It is not clear what the main outcome or confounders. In addition, the sample size is too small to use PS matching. It’s enough to use age- and sex- matching, if it is needed.

Response 1:

The main outcome is comparing the clinical and laboratory features of RS3PE and seronegative RA. RS3PE occurs mostly in elderly men (McCarty, DJ; O'Duffy, JD; Pearson, L .; Hunter, J. Remission seronegative symmetrical synovitis with pitting edema. RS3PE syndrome. JAMA 1985 254, 2763–2767.). The prevalence of malignancy is higher among elderly men ([Cancer statistics]. [Website in Japan]. [Internet. Accessed January 04, 2021.] Available from: https://ganjoho.jp/reg_stat (/statistics/dl/index.html). We considers age and gender are confounding factors. We will change to match age and gender according to your comment (line 102-103).

Point 2:

The third analysis is a kind of case-control analysis to identify if the RS3PE is a risk factor for the development of malignancies. In this analysis, malignancy is an outcome and it should have happened after the development of RS3PE or SNRA. However, the associating malignancy was defined as they were diagnosed within 2 years before or after RS3PE or SNRA diagnosis. The purpose of this study was to compare the clinical characteristics of RS3PE and SNRA, but their conclusion is strange. They concluded that patients who are older, male, with extremity edema, or with RS3PE should be screened for malignancies.

Response 2:

Symptoms often don't appear until the malignancy is in its later stages. It is not clear when the onset of malignancy. So far, It is not clear which is the cause of malignancy and RS3PE or SNRA and which is the result of malignancy and RS3PE or SNRA.

This study cannot prove a causal relationship because it is a retrospective study. However, although the cause and effect are not clear, it is possible that they are associated. Therefore, we change the term to “an associated factor (line 106)” instead of “a risk of malignancy”.

The main outcome is comparing the clinical and laboratory features of RS3PE and seronegative RA. The conclusion will also include the results of the main outcome.

Reviewer 2 Report

Dear authors, 

Thank you for the effort and time you put in providing this registered report. The topic is interesting and might be useful for disstinction of PMR, RA and RS3PE. 

Please find my comments below: 

  1. The idea why that PMR and RS3PE should be considered as RS3PE, or PMR and RA should be considered as RA is not clear for me. I suggest that these groups being considered separately, if there is an overlap between the symptoms and the final diagnosis is not yet established. 
  2. You mentioned that your data are from 2007-2020. Therefore, I am wondering how the diagnosis of PMR coulld be established according to 2012 criteria for those patients were diagnosed before 2012. 
  3. I also suggest not to exclude thoses with paraneoplastic arthritis, as the characteritics of these patietns are also of great interest.
  4. In the method section, I suggest to provide more information on what clinical, paraclinical, imaging data is considered to be gathered.  
  5.   I suggest to define your primary, secondary objectives in the paper.
  6. Lastly, as the data has not been gathered (the concept of registered report), I am just wondering how the authors concluded that some variables are not normally distributed and should be discribed with interquartile range.

Sincerely,    

Author Response

Thank you for your comment.

Point 1:

The idea why that PMR and RS3PE should be considered as RS3PE, or PMR and RA should be considered as RA is not clear for me. I suggest that these groups being considered separately, if there is an overlap between the symptoms and the final diagnosis is not yet established.

Response 1:

According to Kimura's report, PMR and RS3PE should be considered as RS3PE.

 (Kimura, M.; Tokuda, Y.; Oshiawa, H.; Yoshida, K.; Utsunomiya, M.; Kobayashi, T.; Deshpande, G.A.; Matsui, K.; Kishimoto, M. Clinical characteristics of patients with remitting seronegative symmetrical synovitis with pitting edema compared to patients with pure polymyalgia rheumatica. J Rheumatol 2012 39, 148–153.)

To meet the 2010 EULAR / ACR criteria for seronegative RA, a diagnosis cannot be made without more than 11 arthritis, including one small joint.

The 2010 ACR/EULAR criteria for RA Score

Classification criteria for RA (a score of ≥6/10 is needed for classification of a patient as having definite RA)

  1. Joint involvement

 1 large joint                                                                                            0

 2−10 large joints                                                                                  1

 1−3 small joints (with or without involvement of large joints)              2

 4−10 small joints (with or without involvement of large joints)            3

 >10 joints (at least 1 small joint)                                                           5

  1. Serology (at least 1 test result is needed for classification)

 Negative RF and negative ACPA                                                           0

 Low-positive RF or low-positive ACPA                                                 2

 High-positive RF or high-positive ACPA                                               3

  1. Acute-phase reactants (at least 1 test result is needed for classification)

 Normal CRP and normal ESR                                                                0

 Abnormal CRP or normal ESR                                                               1

  1. Duration of symptoms

 <6 weeks                                                                                               0

 ≥6 weeks                                                                                               1

Cutolo et al. reported that in comparison with the patients with RA, the patients with PMR had less frequently arthritis of PIP, MCP and wrist (P<0.001) and the combination of wrist + MCP/PIP or wrist + PIP + MCP was highly suggestive of RA (P<0.001). (Cutolo, M.; Cimmino, MA.; Sulli, A. Polymyalgia rheumatica vs late-onset rheumatoid arthritis. Rheumatology 2009 48, 93–95.)

Caporalia et al. reported that the presence of peripheral synovitis could differentiate patients who will develop RA from those with “true” PMR (Caporalia, R.; Montecuccoa, C.; Episa, O.; Bobbio-Pallavicinia, F.; Maiob, T.; Cimminob, MA. Presenting features of polymyalgia rheumatica (PMR) and rheumatoid arthritis with PMR-like onset: a prospective study. Ann Rheum Dis 2001 60, 1021–1024).

It is common for rheumatologists that patients were diagnosed with RA if they meet the diagnostic criteria for PMR and RA and have peripheral arthritis.

However, we will also analyze patients with “pure RS3PE” and “pure seronegative RA”, except for those who meet the PMR criteria.

The definition is lengthy in “method”, so it will be described in “discussion”.

The following has been added to the text (line 73-74).

We defined the RS3PE and seronegative RA patients excluding the patients who met the criteria for PMR as “pure RS3PE” and “pure seronegative RA” patients.

Point 2:

You mentioned that your data are from 2007-2020. Therefore, I am wondering how the diagnosis of PMR coulld be established according to 2012 criteria for those patients were diagnosed before 2012.

Response 2:

For the patients diagnosed with PMR before 2012,

we retrospectively reviewed the if they met the 2012 PMR classification criteria.

2012 PMR classification criteria scoring algorithm

required criteria: age ≥50 years, bilateral shoulder aching, and abnormal CRP and/or ESR

Points without US (0–6)  

Morning stiffness duration >45 minutes       2           

Hip pain or limited range of motion               1           

Absence of RF or ACPA                                2           

Absence of other joint involvement                1           

A score of 4 or more is categorized as polymyalgia rheumatica (PMR) in the algorithm without ultrasound (US)

We asked our routines about morning stiffness duration at the first visit routinely, so we can retroactively use the 2012 criteria. In seronegative patients with required criteria(age ≥50 years, bilateral shoulder aching, and abnormal CRP and/or ESR), the 2012 PMR criteria will be met if morning stiffness duration> 45 minutes is met.

Point 3:

I also suggest not to exclude thoses with paraneoplastic arthritis, as the characteritics of these patients are also of great interest.

Response 3:

I agree with you.

The patients with paraneoplastic arthritis are not excluded and are evaluated as “the patients with paraneoplastic arthritis”.

We also describe the results of patients with paraneoplastic arthritis.

The following has been added to the manuscript (line 76).

Patients with paraneoplastic polyarthritis were excluded “from patients with RS3PE or seronegative RA”.

Point 4:

In the method section, I suggest to provide more information on what clinical, paraclinical, imaging data is considered to be gathered.

Response 4:

The following has been added to the manuscript (line 95-99).

“Edema was evaluated separately for edema of only hands, only feet, and of both limbs.”

“If there were evaluable examinations, ultrasound imaging, breast imaging, joint X-ray, chest computed tomography (CT), abdominal CT, pelvic CT, positron emission tomography -CT, joint magnetic resonance imaging, upper and lower gastrointestinal endoscopy, gynecological examination, and pathological test were evaluated. ”

Point 5:

I suggest to define your primary, secondary objectives in the paper.

Response 5:

I agree with you.

The following has been added to the manuscript (line 53-56).

This study primarily aimed to compare the clinical characteristics of RS3PE and seronegative RA and evaluate the frequency of concurrent malignancy. This study secondarily aimed to compare the clinical features with and without malignancies in patients with RS3PE, and to compare the clinical features with and without malignancies in patients with seronegative RA.

Point 6:

Lastly, as the data has not been gathered (the concept of registered report), I am just wondering how the authors concluded that some variables are not normally distributed and should be described with interquartile range.

Response 6:

The following has been added to the manuscript (line 107-109).

Results that do not follow the Gaussian distribution are expressed as the median of the 25-75th percentile (interquartile range), and results that follow the Gaussian distribution are expressed as mean ± standard deviation.

Reviewer 3 Report

  1. In introduction the authors described "..... ,there are no reports comparing the characteristics....  . This should be modifided because recently kawashiri et al. reported the difference of these two disease (Clin Rheumatol. 2020 Jun;39(6):1981-1988. ).
  2. If possible the difference of level of serum VEGF should be analyzed.   

Author Response

Thank you for your comment.

Point 1:

In introduction the authors described "..... ,there are no reports comparing the characteristics....  . This should be modifided because recently kawashiri et al. reported the difference of these two disease (Clin Rheumatol. 2020 Jun;39(6):1981-1988. ).

Response 1:

The following has been added to the manuscript (line 43-46).

Kawashiri et al. reported the differences in musculoskeletal ultrasound findings of both hands between RS3PE and “seropositive” elderly-onset RA, however, to our knowledge, there are no reports comparing the characteristics of RS3PE and “seronegative” RA.

(Kawashiri, S.; Suzuki, T.; Okada, A; Tsuji, A.; Takatani, A.; Shimizu, T.; Koga, T.; Iwamoto, N.; Ichinose, K.; Nakamura, H.; Origuchi, T.; Kawakami, A. Differences in musculoskeletal ultrasound findings between RS3PE syndrome and elderly-onset rheumatoid arthritis. Clin Rheumatol 2020 39, 1981-1988.)

Point 2:

If possible the difference of level of serum VEGF should be analyzed.

Response 2:

Unfortunately, there is no data because serum VEGF is not measured in general practice, but we will consider measuring it for research purposes in the future. We will discuss serum VEGF further in the “discussion” section, as there is a paper by Arima et al.

(Arima, K.; Origuchi, T.; Tamai, M.; Iwanaga, N.; Izumi, Y.; Huang, M.; Tanaka, F.; Kamachi, M.; Aratake, K.; Nakamura, H.; Ida, H.; Uetani, M.; Kawakami, A.; Eguchi, K. RS3PE syndrome presenting as vascular endothelial growth factor associated disorder. Ann Rheum Dis 2005 64, 1653–1655. )

Round 2

Reviewer 1 Report

There are some similar or identical phrases in the introduction and methods sections in the manuscript.  English proofreading is required to improve the quality of writing.   

Author Response

Thank you for your comment.

Point 1:

There are some similar or identical phrases in the introduction and methods sections in the manuscript. English proofreading is required to improve the quality of writing. 

Response 1:

We avoided a similar or identical phrase in the introduction and methods sections. (line 32-34). We got English proofreading and corrected our manuscript carefully. The underlined parts are changes.

Reviewer 2 Report

Thank you authors, 

I have no further comments. Looking forward to seeing the results of the study. 

Sincerely, 

Author Response

Thank you for your comment.

We look forward to working with you to move this manuscript closer to publication.

Reviewer 3 Report

The author replied adequately.

Author Response

(The authors gave the same response as above.)
